# Enhanced Mechanic Strength and Thermal Conductivities of Mica Composites with Mimicking Shell Nacre Structure

**DOI:** 10.3390/nano12132155

**Published:** 2022-06-23

**Authors:** Fuqiang Tian, Jinmei Cao, Yiming Li

**Affiliations:** 1School of Electrical Engineering, Beijing Jiaotong University, Beijing 100044, China; 21117019@bjtu.edu.cn; 2State Grid Beijing Electric Power Company Maintenance Branch, Beijing 100089, China; 13269309776@126.com

**Keywords:** shell nacre structure, mica tape, mica composite, thermal conductivity, mechanical property, electrical property

## Abstract

As the main insulation of high-voltage motors, the poor mechanical and thermal conductivities of mica paper restrict the motor’s technological advances. This paper prepared multilayer toughening mica composites with a highly ordered “brick-mud” stacking structure by mimicking the natural conch nacre structure. We investigated the mechanical, thermal, and breakdown properties by combined study of tensile strength, stiffness, thermal conductivity, and breakdown strength at varying mica and nanocellulose contents. The results show that thermal conductivity of the mica/chitosan composites were gradually enhanced with the increase in mica content and the composite shows the optimal synthetic performance at 50 wt% mica content. Further addition of the nanocellulose can extremely enhance the thermal conductivities of mica/chitosan composites. The composite with 5 wt% nanocellulose obtained the maximal thermal conductivity of 0.71 W/(m·K), which was about 1.7 times that of the mica/chitosan composite (0.42 W/(m·K)) and much higher than normal mica tape (0.20 W/(m·K)). Meanwhile, the breakdown strength and tensile strength of mica/chitosan/nanocellulose composite also demonstrated substantial improvement. The application of the mica/chitosan/nanocellulose composite is expected to essentially enhance the stator power density and heat dissipation ability of large-capacity generators and HV electric motors.

## 1. Introduction

Large high-voltage generators are the key equipment for renewable energy power generation such as wind, hydroelectric, and nuclear power generation. As the main insulation for generator stator winding, mica paper has excellent electrical insulation and heat resistance [1,2]. However, with the development of power industry and the increasing power density of high-voltage electric generators and motors, long-term operation of equipment will generate more heat and lead to local overheating of core components, thereby reducing the reliability and life time of the generators and motors [3]. Statistically, nearly 40% of motor faults are caused by the mica tape aging. Heat accumulation and temperature abnormity are the major contributors to insulation deterioration [4]. Studies have shown that a 20% increase in generation efficiency is achieved by increasing the thermal conductivity of the mica tape by 50%, while maintaining the generator volume and voltage level constant. If the thermal conductivity of main insulation raised from 0.3 W/(m·K) to 1.0 W/(m·K), the temperature classification of insulation will improve from F to H and the operation electric filed can be enhanced from 2.5 kV/mm to 4.0 kV/mm, the insulation thickness of stator winding will be reduced by 40%. Therefore, it is of great significance to the develop high-voltage generators and motors with large capacity and high-power density by improving the thermal conductivity and thinning thickness of the main insulating mica tape.

Mica tape for high-voltage motors has gone through a development process from a double-sided reinforcing material (three-layer structure) to a single-sided glass cloth reinforcing (two-layer structure), which mainly consists of mica paper, polymer adhesive and reinforced material (polyester film or glass fiber cloth). Researchers mainly improve the heat dissipation of the mica tape via the following two aspects [5]. The first way is the preparation of high thermal conductivity mica tape through filling inorganic fillers such as BN, AlN, Al_2_O_3_, etc. in mica paper or adhesive. However, the effect seems limited and the mechanical properties are sacrificed [6,7,8,9]. The second approach is to reduce the mica tape thickness through lowering the addition of adhesive or finding alternative reinforced material. Recent works have reported some new inorganic-polymer composite insulating materials with improved mechanical performances, which is inspired by the structure of natural conch nacre [10,11,12]. This provides an inspiration for designing high-strength mica composites without reinforced materials [13,14,15]. Chitosan has higher crystallization and greater application potential than cellulose due to its excellent film-forming property as a functional biopolymer [16]. Furthermore, the free hydroxyl and amino groups are widely distributed in the surface of chitosan beneficial to mechanical reinforcement of the developing composites [17]. This paper aims to remove the reinforcement layer and design a high-strength mica-based composite insulation material without reinforcement layer (single-layer structure) by modifying the mica paper. The composite material with this structure will have higher mica content than existing mica tapes and better breakdown field strength and electrical aging resistance, which helps to thin the main insulation thickness and increase the motor stator power density.

In this paper, the multilayer toughening mica composites with highly ordered “brick-mud” stacking structure was prepared by mimicking the natural shell nacre structure. The mechanical, thermal, and electrical insulation properties of composites were investigated.

## 2. Materials and Methods

### 2.1. Main Materials

Mica powder (MF1250) was purchased from Baofeng Mica Processing Co., Ltd., Shijiazhuang, China. Silane coupling agent (JFCG5), nanocellulose and glycerol was provided by Suzhou Jufeng electrical insulation system Limited by Share Ltd., Suzhou, China. Chitosan with the viscosity of 1.40 × 10^8^ mPa.s was purchased from Saien Biological Technology Co., Ltd., Xian, China. Acetic acid with 99.5% purity was purchased from Xilong Scientific Co., Ltd., Guangzhou, China.

### 2.2. Fabrication of Mica Composites

Modified mica powder were prepared by adding JFCG5 and mica powder in the proportion 1:99 to high-speed mixer (BS-65) under constant stirring at 90 °C for 1 h, followed by coupling reaction under 150 °C for 12 h and stirred for 5 min. Chitosan solution was prepared by mixing the acetic acid, chitosan, glycerol, and water in the proportion 1:4:2:100 under stirring and dispersing in a homogenizer for 2 h. Mica composites was fabricated by adding modified mica powder and cellulose in varying proportions to chitosan solution under stirring ang dispersing in a homogenizer for 5 h, degassed sufficiently in a vacuum drying oven for 30 min at room temperature, coated on the substrate and cured under 70 °C for 8 h.

### 2.3. Characterization

Cross-section morphologies of the samples was observed under high resolution field emission scanning electron microscope (SU8020, Hitachi, Japan) operated at 10 kV. The tensile strength and elongation at break of mica composites were measured by universal testing machine (UTM2502, Jufeng, China) at a tensile rate of 10 mm/min. The specimen size was 250 mm × 20 mm × 0.1 mm. The thermal conductivity of all samples were measured through a thermal conductivity measurer (DRL-III, Xiangyi, China). The dielectric breakdown strength of all samples were tested by a withstand voltage tester (CW2672H, Jufeng, China) at least 10 samples.

## 3. Results and Discussion

### 3.1. Subsection Nanoparticle Dispersion Characterization

Figure 1 shows the cross-section of mica/chitosan and mica/chitosan/nanocellulose composites with different mica and nanocellulose mass fraction. From Figure 1a–c, the multilayer toughened stacking structures similar to the natural shell nacre structure gradually appear in the composite with the increase of mica content. As shown in Figure 1d, the mica will agglomerate into clusters by further increasing the content in the composite. Furthermore, the conditions of phase separation and microvoids will be generated, which will result in a reduction in bulk density and a decrease in the properties of the composite. The bulk density of the mica/chitosan composites reached a maximum at the mica content of 50 wt%. On this basis, nanocellulose was added to the composites. As shown in Figure 1e,f, the cross-section of the composite is fluffy and rough at a nanocellulose content of 15 wt%, and there is a large amount of agglomerated nanocellulose between the mica sheets. This can be attributed to the weak bond strength between the mica and chitosan. The addition of 5 wt% of nanocellulose produced more interfacial adhesion in the composite.

### 3.2. Mechanics Performance Analysis

Figure 2 shows the tensile strength, elongation at break, and stiffness of the mica/chitosan and mica/chitosan/cellulose composites. In Figure 2a, with the increasing of mica content, the tensile strength and elongation at break of mica/chitosan composite increases first, and then decreases. The composite exhibits excellent mechanical properties at an optimum ratio of 1:1 of mica to chitosan particles. Furthermore, the tensile strength and elongation at break of the composite increased by 56.40% (78.20 N/10 mm) and 87.90% (10.90%), respectively. This is due to the fact that chitosan molecular chains intertwine during film formation to form a low regularity material due to the high relative molecular weight and long molecular chains, while mica and chitosan can interact with each other and gradually form a close stacking structure. The dense structure of the composite improves with the mica content, which shows a gradual increase in tensile strength and elongation at break. The accumulation of too much mica in the chitosan will weaken the interaction between the mica and chitosan particles, leading to the separation of the mica and chitosan phases when the mica content reaches a threshold. In addition, air bubbles are introduced at the interface between the two phases with the further increase of mica content, thus the tensile strength and elongation at break gradually decrease. As a result, mica and chitosan build up a tightly packed structure in the composite, and the best insulation properties are achieved when the mica content is around 50 wt%. The stiffness of mica tape is generally less than 50 N/m and is a very important indicator that directly affects the finished product. In Figure 2b, the stiffness of the composite increases gradually with the mica content, and the sample with 80% mica content breaks during the measurement with a maximum stiffness of 41.1 N/m. Therefore, the sample performance meets the insulation requirements of mica tape for high voltage motors.

The mechanical and electrical breakdown properties must be maintained while improving the thermal conductivity of the composite. Based on the above analysis, nanocellulose was added to the mica/chitosan composite with the mica content maintained at 50 wt% and the measured tensile strength of the composite is shown in Figure 2c. The addition of a small amount of nanocellulose can greatly improve the tensile strength of the composite. The tensile strength of the composite increases by 27.30% (99.60 N/10 mm) at 5 wt% nanocellulose content, and when the nanocellulose is increased to 15 wt%, the tensile strength of the composite insulation is 122.20 N/10 mm (i.e., increased by 56.30%). The crosslinking between the nanocellulose and chitosan can enhance the interaction between molecular chains in the composite [18]. In addition, the nanocellulose is able to fill the voids between the mica and chitosan due to its small size effect (50–200 nm) and high mechanical strength, and further increases the overall density of the composite insulation. Thus, the tensile strength of the composite increases with the nanocellulose content.

### 3.3. Thermal Performance Analysis

Thermal conductivity of the mica/chitosan and mica/chitosan/nanocellulose composites are shown in Figure 3. In Figure 3a, the thermal conductivity of the mica/chitosan composites is not significantly improved when the mica content is less than 40 wt%, which only increases from about 0.31 W/(m·K) at 20 wt% to about 0.36 W/(m·K) at 40 wt% mica content of composite, while the thermal conductivity of pure chitosan is about 0.3 W/(m·K). This happened due to the fact that the mica will be coated by chitosan and forms a “sea-island” system when the mica filling is less, the thermal conductivity of composite at this time is mainly influenced by the interfacial thermal resistance between mica and chitosan. The thermal conductivity of the composites increases significantly due to mica sheets overlap and interact with each other to form hydrogen bonds with the further increase of mica content, which seriously decrease the interfacial thermal resistance of composites, so the thermal conductivity rises by 70% (0.51 W/(m·K)) when the mica content reaches 80 wt%.

In Figure 3b, the thermal conductivity of mica/chitosan/nanocellulose increases first and then decreases with nanocellulose content. The addition of trace amounts of nanocellulose to the mica/chitosan composites resulted in better heat dissipation capability with a maximum thermal conductivity of 0.71 W/(m·K), which is 355% higher than the mica tape product in Table 1. Heat transfer in polymers is generally accomplished through molecular chains, and nanocellulose can be cross-linked with chitosan molecular chains and gradually form a local thermal conductivity network in composite due to its large specific surface area. Therefore, the thermal conductivity of composite insulation material is significantly improved with increasing nanocellulose content. Nanocellulose is easy to agglomerate when its content exceeds the threshold value in composite, which seriously increases the contact area of each phase and then leads to an increase in the interfacial thermal resistance, and thus the thermal conductivity began to decrease at nanocellulose content above 5 wt%.

### 3.4. Breakdown Performance Analysis

Figure 4a shows the Weibull distribution of AC breakdown field strength of mica/chitosan composites with different mica content, and the breakdown field strength is obtained with a probability of 63.20%, as shown in Figure 4b. With the addition of mica, the breakdown field strength of the composites increased continuously, with the highest increase being 39.5% for a mica content of 70 wt%. However, the breakdown field strength suddenly decreased to 10.62 kV/mm with the further increase of mica amount. This happened due to the fact that excess mica aggregates together in the chitosan solution and then the interaction between mica and chitosan particles is weakened and two-phase separation occurs. In addition, the composite material structure became loose and had more defects due to the small thickness (0.10 mm~0.15 mm) of the composite material and the large mica particle size when filled with too much mica. The initial amount of electrons inside the material increases under the action of external electric field, and more electrons are generated by collisions between electrons, leading to a decrease in breakdown strength. Seen from the Figure 4c and Table 1, the breakdown strength of the composite material remains between 18 kV/mm and 19 kV/mm at thinner thicknesses (0.12 mm), which is better than the national standard of 15 kV/mm for mica tape. Therefore, the addition of nanocellulose in the composite can improves the thermal conductivity and mechanical properties without affecting its breakdown strength.

## 4. Conclusions

Multilayer toughening mica composites with a highly ordered “brick-mud” stacking structure was prepared by mimicking the natural shell nacre structure. The mechanical, thermal, and electrical insulation properties of composites were investigated via measurement of tensile strength, stiffness, thermal conductivity, and breakdown strength at varying mica and nanocellulose content. The research suggests that the thermal conductivity of mica/chitosan composites were gradually enhanced with the increase in mica content and the composite shows the optimal synthetic performance at 50 wt% mica content. Further addition of the nanocellulose can extremely enhance the thermal conductivities of mica/chitosan composites. Meantime, the breakdown strength and tensile strength of mica/chitosan/nanocellulose composite also got substantially improvement. In summary, all of the properties of the mica/chitosan/nanocellulose composite are better than the mica tapes on the market today, which is expected to essentially enhance the stator power density and heat dissipation ability of large-capacity generators and HV electric motors.

## Figures and Tables

**Figure 1 nanomaterials-12-02155-f001:**
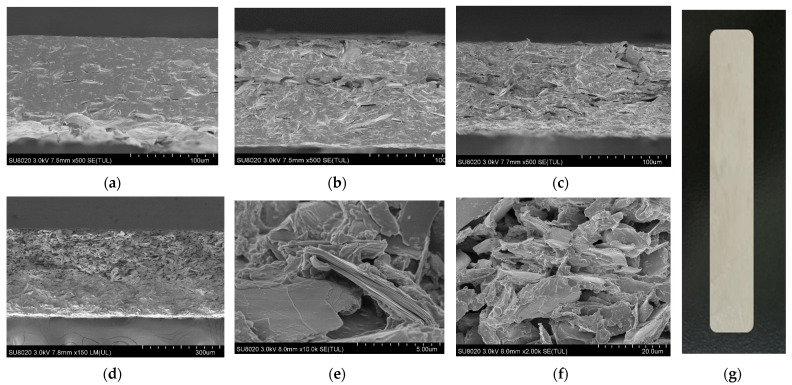
Cross-sectional images of the mica/chitosan composites with different mica mass fraction: (**a**) 20 wt%; (**b**) 40 wt%; (**c**) 50 wt%; (**d**) 70 wt%. Cross-section images of the mica/chitosan/nanocellulose composites with different nanocellulose contents: (**e**) 5 wt%; (**f**) 15 wt%, the mass fraction of mica is controlled to be 50 wt%. (**g**) The samples of mica/chitosan composites.

**Figure 2 nanomaterials-12-02155-f002:**
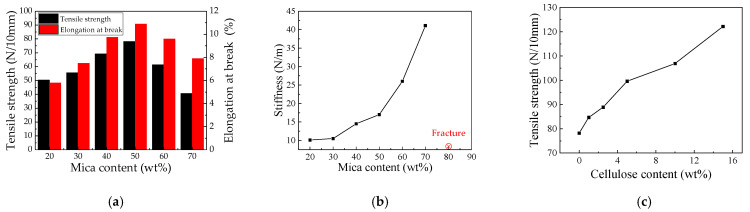
Mechanics properties of the mica composites (**a**,**b**) and mica/chitosan/nanocellulose composites (**c**).

**Figure 3 nanomaterials-12-02155-f003:**
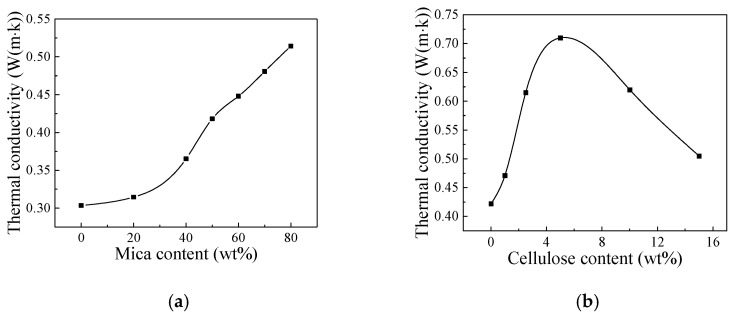
Thermal conductivity of the mica composites (**a**) and mica/chitosan/nanocelluloses composites (**b**).

**Figure 4 nanomaterials-12-02155-f004:**
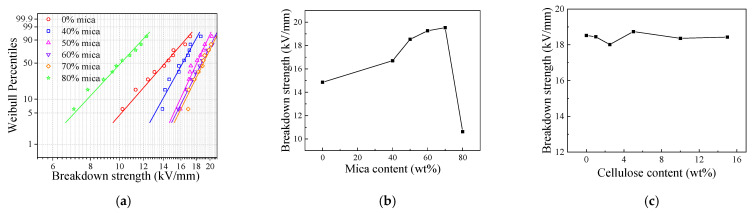
Breakdown strength of the mica composites (**a**,**b**) and mica/chitosan/nanocellulose composites (**c**).

**Table 1 nanomaterials-12-02155-t001:** The performance table of the mica/chitosan and mica/chitosan/nanocellulose composites.

	50% Mica + 5% Nanocellulose	50% Mica	70% Mica	GB/T 5019.12-2017
Thickness/mm	0.12 ± 0.02	0.12 ± 0.02	0.15 ± 0.02	0.15 ± 0.02
Total weight/(g/m^2^)	154.34	149.14	238.69	215 ± 20
Mica content/(g/m^2^)	74.68	74.57	167.1	180 ± 20
Glass fabric/(g/m^2^)	0	0	0	23 ± 2
Accelerator content/(g/m^2^)	74.68	74.57	71.61	12 ± 4
Tensile strength/(N/10 mm)	99.60	78.20	40.70	≥80
Breakdown strength/(kV/mm)	18.64	18.53	19.53	≥15
Bulk resistivity (Ω·m)	10^10^	10^10^	10^10^	-
Thermal conductivity/(W/(m·K))	0.71	0.42	0.48	0.20
Stiffness (N/m)	17	17	41.1	≤50

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
