# Peer review of "Enhanced Mechanic Strength and Thermal Conductivities of Mica Composites with Mimicking Shell Nacre Structure"

_nanomaterials, 2022, doi:10.3390/nano12132155_

Round 1
Reviewer 1 Report
this is a largely phenomalogical paper in which a series of composites were produced and subsequently tested for mechanical properties and thermal conductivity. There is little information in the paper to provide any fundamental understanding of the loading and morphology that is generated which diminishes the value of the paper. For example, it would be preferable to use volume % rather than weight percent to understand the interaction of the particle and the thermal properties.
The morphology for the higher loadings of mica appears to be very different to lower loadings which impacts the mechanical properties in particular.
I don't believe that the composites are well formed and may not be uniformly formed based on the information in the paper
Author Response
Dear Editor,
We would like to thank you for your kind letter and all the reviewers’ constructive comments. These comments are all valuable and helpful for improving this manuscript entitled “Enhanced mechanic strength and thermal conductivities of mica composites with mimicking shell nacre structure (ID: nanomaterials-1777643)”. According to the reviewers’ comments, we have modified the manuscript to meet the requirements. Modifications are marked with red color in the manuscript. Point-by-point responses to the reviewers are listed below in this letter. We would like to express our sincere thanks to the reviewers and you for the constructive and positive comments.
Point 1: This is a largely phenomalogical paper in which a series of composites were produced and subsequently tested for mechanical properties and thermal conductivity. There is little information in the paper to provide any fundamental understanding of the loading and morphology that is generated which diminishes the value of the paper. For example, it would be preferable to use volume % rather than weight percent to understand the interaction of the particle and the thermal properties.
Response 1: The application of mica paper as the main insulation materials in high voltage motors requires the use of glass cloth for reinforcement to form mica tape due to the poor mechanical properties of mica paper. The including of glass cloth will decrease the insulation properties and thermal conductivity of the mica tape. We aim to design mica-based composite with excellent thermal, insulating and mechanical properties but without glass cloth reinforcement. This attempt is a major innovation for high voltage motor insulaiton and will lead to a new generation main insulation materials, which has great significance for high voltage and large capacity generator and motor development. The mica mass is one of the criteria to measure the performance of mica tape, so we use the mass ratio rather than volume ratio in order to make a comparison between mica tape and the mica based composite in our manuscript. In addition, the fundamental understanding of the reults has been described in the “results and discussion” part.
Point 2: The morphology for the higher loadings of mica appears to be very different to lower loadings which impacts the mechanical properties in particular.
Response 2: The microscopic morphology and properties of the composite do change dramatically as the mica content increases. Just as shown in Figure 2 and 3, the agglomeration of mica in the chitosan weakens the interaction between the mica and chitosan particles when the mica content reaches a threshold, leading to the phase separation of the mica and chitosan (Figure 2(d)), which will result in the poor tensile strength and elongation at break.
Point 3: I don't believe that the composites are well formed and may not be uniformly formed based on the information in the paper.
Response 3: As explained in “Response 2”, the microscopic morphology and properties of the composite do change dramatically as the mica content increases. Just as shown in Figure 2 and 3, the agglomeration of mica in the chitosan weakens the interaction between the mica and chitosan particles when the mica content reaches a threshold, leading to the phase separation of the mica and chitosan (Figure 2(d)), which will result in the poor tensile strength and elongation at break. However, as shown in Figure 2, the composites with mica content less than 50wt% shows good dispersion and regular morphology structure, whch is also in good agreeement with the significant enhancement of the mechanical and electrical breakdown properties. The samples of the mica/chitosan composite has been supplemented in Figure 2(g), it can be seen that the surface of our prepared composites is smooth and flat.
If you have any queries, please don’t hesitate to contact us.
Thank you and best regards.
Yours sincerely,
Jinmei Cao

Reviewer 2 Report
The research of Tian et al, as well reflected by the title of this manuscript, is focused on mica-based composites with a structure inspired by nature and able to guarantee an improvement in the current generation efficiency of high-voltage motors.
Specifically, the authors consider mica/chitosan/nanocellulose composites with a multilayer structure similar to that of natural shells.
All the formulations, prepared with a simple and clearly described methodology, have been systematically characterized in terms of morphological aspects, thermal conductivity, mechanical tensile performance and AC breakdown strength, using suitable techniques.
The results achieved by varying the composition of the composites are clearly reported and properly discussed also with the support of appropriate graphic representations.
The study is interesting and relevant given the current interest in alternative energy sources for which high-power generators play a key role. The in-depth knowledge generated by this research and easily transferable to interested readers, even if unfamiliar with the aspects covered, given the clarity of presentation of the contents, is appreciable.
The manuscript is suitable for publication.
Author Response
Dear Editor,
We would like to thank you for your kind letter and all the reviewers’ constructive comments. These comments are all valuable and helpful for improving this manuscript entitled “Enhanced mechanic strength and thermal conductivities of mica composites with mimicking shell nacre structure (ID: nanomaterials-1777643)”. We would like to express our sincere thanks to the reviewers and you for the constructive and positive comments.
If you have any queries, please don’t hesitate to contact us.
Thank you and best regards.
Yours sincerely,
Jinmei Cao

Reviewer 3 Report
Title: “Enhanced mechanic strength and thermal conductivities of mica composites with mimicking shell nacre structure”
Journal: Nanomaterials
Manuscript Number: nanomaterials-1777643
Tian et al. examined the mechanical, thermal, and breakdown properties of mica and nanocellulose composites by studying stiffness, tensile strength, thermal conductivity, and strength at varying amounts of mica and nanocellulose. According to the results, the thermal conductivity of the mica/chitosan composites has improved with an increase in mica content. The composite has optimal performance at a mica content of 50 wt%. Despite this, I believe that the current paper is lacking in novelty. Therefore, this article should only be published with extensive, significant revisions. It is necessary for the authors to devote a few paragraphs to describe the novelty and additional experiments of the following publication.
There are few points the author needs to address:
- Please take the time to read all the papers concerning the mechanical strength and thermal conductivity of mica composites with mimicking shells. A number of publications have been published in this context. Could you please explain the difference between your work and others?
- I recommend that the Introduction, results, and conclusion be revised.
- In order to better explain the concept and process, it is essential to conduct the DSC and TGA analysis of different mica and nanocellulose content.
- I am interested in learning more about the chemistry behind mica and nanocellulose content.
- The authors are expected to provide high-resolution SEM images. SEM images cover very small cross-sectional areas. As a result, it is not possible to determine the overall picture.
- Have the authors checked the storage modulus and loss modulus with temperature or frequency sweeps?
- Scientific insight is lacking in the manuscript. Therefore, additional information should be provided.
Author Response
Dear Editor,
We would like to thank you for your kind letter and all the reviewers’ constructive comments. These comments are all valuable and helpful for improving this manuscript entitled “Enhanced mechanic strength and thermal conductivities of mica composites with mimicking shell nacre structure (ID: nanomaterials-1777643)”. According to the reviewers’ comments, we have modified the manuscript to meet the requirements. Modifications are marked with red color in the manuscript. Point-by-point responses to the reviewers are listed below in this letter. We would like to express our sincere thanks to the reviewers and you for the constructive and positive comments.
Tian et al. examined the mechanical, thermal, and breakdown properties of mica and nanocellulose composites by studying stiffness, tensile strength, thermal conductivity, and strength at varying amounts of mica and nanocellulose. According to the results, the thermal conductivity of the mica/chitosan composites has improved with an increase in mica content. The composite has optimal performance at a mica content of 50 wt%. Despite this, I believe that the current paper is lacking in novelty. Therefore, this article should only be published with extensive, significant revisions. It is necessary for the authors to devote a few paragraphs to describe the novelty and additional experiments of the following publication.
There are few points the author needs to address:
Point 1: Please take the time to read all the papers concerning the mechanical strength and thermal conductivity of mica composites with mimicking shells. A number of publications have been published in this context. Could you please explain the difference between your work and others?
Response 1: The application of mica paper as the main insulation materials in high voltage motors requires the use of glass cloth for reinforcement to form mica tape due to the poor mechanical properties of mica paper. Mica tape for high-voltage motors has gone through a development process from a double-sided reinforcing material (three-layer structure) to a single-sided glass cloth reinforcing (two-layer structure, Figure 1) till now. The including of glass cloth will decrease the insulation properties and thermal conductivity of the mica tape. We aim to design mica-based composite with excellent thermal, insulating and mechanical properties but without glass cloth reinforcement. This attempt is a major innovation for high voltage motor insulaiton and will lead to a new generation main insulation materials, which has great significance for high voltage and large capacity generator and motor development. The composite material with this structure has higher mica content than existing mica tapes and better electrical breakdown strength and electrical aging life, which helps to thin the main insulation thickness and increase the motor stator power density. The published mica/aramid paper composites with simulated shell structure have much less mica content than our present research, which lead to poor electrical insulating properties and shorter electrical aging lifte. In addition, the published research mainly focus on enhancing the mechanical properties and pay less attention to the thermally conductive properties, our present research intends to enhance the thermal properties, electrical insulating properties and mechanical properties concurrently by constructing mimicking shell nacre structure, enhancing mica content and including nanocellulose. The increase in the thermal conductivity of the mica composite is of great significance for heat dissipation and enhancing the power density of high voltage generators and motors.
Point 2: I recommend that the Introduction, results, and conclusion be revised.
Response 2:
We made several amendments to the “Introduction” as shown below.
“Mica tape for high-voltage motors has gone through a development process from a double-sided reinforcing material (three-layer structure) to a single-sided glass cloth reinforcing (two-layer structure), which mainly consists of mica paper, polymer adhesive and reinforced material (polyester film or glass fiber cloth). Researchers mainly improve the heat dissipation of the mica tape via the following two aspects [5]. The first way is the preparation of high thermal conductivity mica tape through filling inorganic fillers such as BN, AlN, Al2O3, ect in mica paper or adhesive, but the effect seems limited and the mechanical properties are sacrificed [6-9]. The second approach is to reduce the mica tape thickness through lower the addition of adhesive or finding alternative reinforced material. Recent works reported some new inorganic-polymer composite insulating materials with improved mechanical performances, which inspired by the structure of natural conch nacre [10-12]. This provides an inspiration for designing high-strength mica composites without reinforced materials [13-15]. Chitosan has higher crystallization and greater application potential than cellulose due to its excellent film-forming property as a functional biopolymer [16]. Furthermore, the free hydroxyl and amino groups are widely distributed in the surface of chitosan beneficial to mechanical reinforcement of the developing composites [17]. This paper aims to remove the reinforcement layer and design a high-strength mica-based composite insulation material without reinforcement layer (single-layer structure) by modifying the mica paper. The composite material with this structure will have higher mica content than existing mica tapes and better breakdown field strength and electrical aging resistance, which helps to thin the main insulation thickness and increase the motor stator power density.”
We made several amendments to the Figure 1, shown below.
Reviewer 4 Report
The authors studied composites containing Mica microparticles (without any indication on the particle size), in combination with chitosan and nanocellulose.
The final material is a microcomposite, and I don't think that this paper corresponds to the journal topic, which is NANOMATERIALS.
This is the reason why I don't think that this paper is adapted for the journal, and the authors should submit it to another journal, mainly focusing on mineral materials or ceramics.
Author Response
Dear Editor,
We would like to thank you for your kind letter and all the reviewers’ constructive comments. These comments are all valuable and helpful for improving this manuscript entitled “Enhanced mechanic strength and thermal conductivities of mica composites with mimicking shell nacre structure (ID: nanomaterials-1777643)”. We would like to express our sincere thanks to the reviewers and you for the constructive and positive comments.
Thank you and best regards.
Yours sincerely,
Jinmei Cao
Round 2
Reviewer 3 Report
Title: “Enhanced mechanic strength and thermal conductivities of mica composites with mimicking shell nacre structure”
Journal: Nanomaterials
Number of the manuscript: nanomaterials-1777643
The current format of the articles needs to be modified and revised. It appears that the author has not answered all of the questions I raised in my first revision. Following the addition of all points to the paper, the editor can accept this manuscript for publication if they believe that everything is in order.
The author needs to address the following points:
The introduction, results, and conclusion should be revised. However, significant improvements are not required.
To determine mica and nanocellulose content, it is necessary to perform both DSC and TGA analyses. Please include in the analysis and correlate the chemistry behind mica and nanocellulose content.
It is expected that the authors will provide high-resolution SEM images. The cross-sections covered by the SEM images are extremely small. Thus, it is impossible to determine the overall picture.
Have the authors checked the storage modulus and the loss modulus using temperature or frequency sweeps?
In the manuscript, there is a lack of scientific insight. Therefore, additional information is required.
Author Response
Response to Reviewer 4 Comments
Dear Editor,
We would like to thank you for your kind letter and all the reviewers’ constructive comments. These comments are all valuable and helpful for improving this manuscript entitled “Enhanced mechanic strength and thermal conductivities of mica composites with mimicking shell nacre structure (ID: nanomaterials-1777643)”. According to the reviewers’ comments, we have modified the manuscript to meet the requirements. Modifications are marked with red color in the manuscript. Point-by-point responses to the reviewers are listed below in this letter. We would like to express our sincere thanks to the reviewers and you for the constructive and positive comments.
The current format of the articles needs to be modified and revised. It appears that the author has not answered all of the questions I raised in my first revision. Following the addition of all points to the paper, the editor can accept this manuscript for publication if they believe that everything is in order.
The author needs to address the following points:
Point 1: The introduction, results, and conclusion should be revised. However, significant improvements are not required.
Response 1: We made several amendments to the introduction, results and conclusion, shown in manuscript.
Point 2: To determine mica and nanocellulose content, it is necessary to perform both DSC and TGA analyses. Please include in the analysis and correlate the chemistry behind mica and nanocellulose content.
Response 2:
The content of mica and nanocellulose has been specified during the preparation of the composites, so the maesurement of TG is not necessary.
The purpose of adding nanocellulose to the composite is to enhance interfacial adhesion between mica and chitosan and to form a local thermal conductivity network, and the interaction among the mica, chitosan and nanocellulose mainly relies on hydrogen bonding and van der Waals forces. In addition, as an inorganic mineral material with good heat resistance, mica won’t decpmpose when tested at high temperature. Therefore, the heat absorption and exotherm results in the DSC test results of the composites at high mica fillings mainly reflect the thermal decomposition of chitosan and nanocellulose themselves, which is not meaningful.
In summary, TG and DSC analysis is not so necessary in our research for exploring the relationship between composite stucutre and properties.
Point 3: It is expected that the authors will provide high-resolution SEM images. The cross-sections covered by the SEM images are extremely small. Thus, it is impossible to determine the overall picture.
Response 3: We have provided high-resolution SEM images in Figure 1.
Point 4: Have the authors checked the storage modulus and the loss modulus using temperature or frequency sweeps?
Response 4: The main performance parameters for evaluating mica types to meet the national standard and application are thermal conductivity, stiffness, tensile strength and breakdown voltage. Therefore we didn’t measure the storage modulus and the loss modulus of the composites.
Point 5: In the manuscript, there is a lack of scientific insight. Therefore, additional information is required.
Response 5: This paper aims to remove the reinforcement layer and design a high-strength mica-based composite insulation material without reinforcement layer (single-layer structure) by modifying the mica paper. The composite material with this structure will have higher mica content than existing mica tapes and better breakdown field strength and electrical aging resistance, which helps to thin the main insulation thickness and essentially enhance the stator power density and heat dissipation ability of large-capacity generators and HV electric motors.
If you have any queries, please don’t hesitate to contact us.
Thank you and best regards.
Yours sincerely,
Jinmei Cao
